# Treatment of Harvest Mite Infestation in Dogs Using a Permethrin 54.5% and Fipronil 6.1% (Effitix^®^) Topical Spot-On Formulation

**DOI:** 10.3390/vetsci6040100

**Published:** 2019-12-07

**Authors:** Line-Alice Lecru, Daniel Combarros, Eloy Castilla-Castaño, Christelle Navarro, Marie Christine Cadiergues

**Affiliations:** 1Université de Toulouse, ENVT, Small Animal Hospital, Dermatology Service, 23, Chemin des Capelles, CEDEX 3, 31076 Toulouse, France; l.lecru@envt.fr (L.-A.L.); d.combarros@envt.fr (D.C.); eloycascas@gmail.com (E.C.-C.); 2Global Medical and Marketing, Virbac, 13ème rue LID, 06511 Carros, France; christelle.navarro@virbac.com; 3UDEAR, Université de Toulouse, INSERM, ENVT, 23, Chemin des Capelles, CEDEX 3, 31076 Toulouse, France

**Keywords:** dogs, dermatology, trombiculidae, antiparasitic agents

## Abstract

Background—The study aims to assess the efficacy of a permethrin 54.5%-fipronil 6.1%-based spot-on solution in dogs naturally infested with *Neotrombicula* in an open-label controlled study. Methods—Ten naturally infested dogs received one drop per affected site on day (D) 0, and on D14, the rest of the pipette was applied on the skin between the shoulders. Five dogs served as sentinels. Parasite score (0–3), skin lesions (0–4), and investigator pruritus scale (0–4) were assessed on D0, D1, D14, and D28. Results—No treated dogs developed adverse reactions. Parasite score of sentinel dogs was maintained between 1.8 (D0, D1, and D28) and 2.2 (D14). In treated dogs, D0 parasite score was 2.4. It was significantly reduced from D1 (0.5; *p* < 0.002) to D28 (0.1; *p* < 0.002). The lesion score was 2.9 on D0 and D1; it was significantly reduced on D14 (0.6; *p* < 0.002) and D28 (0.1; *p* < 0.002). Similarly, investigator pruritus scale (D0, 2.2) scores significantly decreased on D14 (0.4; *p* < 0.004) and D28 (0.2; *p* < 0.002). Conclusions—The combination permethrin-fipronil appears to be well-tolerated, rapidly and durably effective in the control of localized canine harvest mite infestation.

## 1. Introduction

Harvest mites, also known as chigger mites, are common skin parasites that are present worldwide. Harvest mites feed on small rodents, although they are capable of feeding on humans and practically on all species of domestic animals, including birds [1,2,3,4]. They have a strong seasonal activity and are usually found in the highest numbers on hosts in late summer and autumn. They prefer chalky soils to clay and can be numerous on grassland, cornfields, heathland, and scrubby woodland [5]. *Neotrombicula autumnalis* (Acari: Trombiculidae) is considered as the most frequent, but the diversity of harvest mites causing trombiculiasis of domestic animals and humans in Europe is likely underestimated [6]. *N. autumnalis* is a small reddish-orange mite of about 0.7 mm long. Eggs are laid in the soil and around vegetation. Larvae hatch from the eggs and crawl onto a host animal. Only larvae can infest hosts from the outdoor environment [7]. They attach themselves, feed on fluids in the tissue for several days, and then leave the host. They molt into a nymph stage before the adults emerge. The life cycle is completed in 50–55 days. In the northern hemisphere, there is one, possibly two, generation(s) per year; in southern parts of the northern hemisphere, the mites can reproduce all year long [1,7].

They are mainly found on the head, ear margins, eyelids, and interdigital spaces of pets and can be found on the face and limbs of grazing animals, depending upon the host height [1,3]. The larvae insert their mouthparts into the host’s skin and inject cytolytic enzymes. They then feed on partly digested host tissue causing potential irritation and possibly a hypersensitivity reaction. However, high levels of infestation can be encountered without evidence of lesions or signs of discomfort [1,3]. In dogs who demonstrate pruritus caused by harvest mites, erythema, papules, occasional vesicles, and eventually crusts can be observed [3]. Their distribution depends on the access of the larvae to the skin; mites attach themselves without moving much at the body surface. Prosl et al. have reported two Yorkshire terriers massively infested (more than 2000 larvae) cases with neurologic signs [8]. Additionally, harvest mites are potential vectors of infectious diseases. In Asia, they are a well-known vector of ricketsiae, *Orientia tsutsugamushi*. *Anaplasma* and *Borrelia* have been detected in unengorged larvae and in larvae collected on birds, leading to a suspected transstadial passage of these pathogens [9,10,11]. Because of the potential clinical relevance and transmission of diverse infectious agents, it is important to develop efficient protocols to treat and prevent harvest mites infestations.

There is limited data on the efficacy of acaricidal products against *Neotrombicula*, and products licensed specifically for this purpose are sparse. Reports of treatment in pets are limited to field studies of natural infestations that are mostly non-blinded and lacking control groups. Current recommendations consist of using fipronil spray applied once monthly [12], especially on feet and ventrally or permethrin/pyriproxyfen spot-on or spray [13].

The objective of the study is to evaluate the efficacy of a permethrin 54.5%-fipronil 6.1%-based spot-on solution (Effitix^®^; Virbac, Carros, France) in dogs naturally infested with harvest mites.

## 2. Materials and Methods

The study was an open-label controlled study and a one-month follow-up. Fifteen client-owned dogs having regular outdoor access were recruited. Both owners’ written consent and approval from the University (Toulouse Veterinary School, France) Ethical Committee were obtained prior to beginning the study. All diagnostic and therapeutic procedures were performed by licensed veterinarians in the course of routine veterinary health management. Inclusion criteria were the infestation by harvest mites on day 0 and the absence of change in the dogs’ lifestyle throughout the study. Control dogs living in different households, but in the same geographical areas were selected with attached harvest mites but with few or no skin lesions and received a rescue treatment at the end of the study. Exclusion criteria included (i) dogs with systemic illness or condition which could deteriorate during the study period, (ii) dogs with a known sensitivity to fipronil or permethrin, (iii) dogs having received an acaricide during the twelve weeks prior to inclusion.

Four visits were planned. Parasitological and dermatological evaluations were conducted on the day of inclusion (day 0), day 1, day 14 (±2 days), and day 28 (±2 days) for all treated dogs; control dogs were only assessed for a parasitic score. Each case was evaluated by the same investigator on all four visits.

Parasite score (PS) was assessed with a handheld magnifying glass, on a 0–3 scale (0 = absence of harvest mites; 1 = mild, 1–5 harvest mites and only one area of the animal’s body was affected; 2 = moderate, 6–10 harvest mites over one region, or 1–5 harvest mites on more than one region of the animal’s body; 3 = severe, >10 harvest mites/region and more than one region of the animal’s body). Skin lesions with attached parasites were scored on a 0–4 scale (lesion score; LS, 0 = absence of lesion; 1 = very mild: few (<10) crusted/erythematous papules in one body region; 2 = mild: many (>10) crusted/erythematous papules in one body region; 3 = moderate: few (<10) crusted/erythematous papules in more than one body region; 4 = severe: many (>10) crusted/erythematous papules in more than one body region).

The investigator scored the severity of pruritus over 5 min in the examination room on a 0–4 scale (investigator pruritus score; IPS, 0 = the dog was comfortable, grooming like any normal dog; 1 = the dog was grooming, but it was tolerable and the dog remained calm; 2 = the dog was grooming, but it was generally tolerable, although showing occasional signs of agitation; 3 = the dog was grooming quite often, the dog was uncomfortable, nervous, or often agitated; 4 = the dog was uncomfortable, grooming all the time) on each visit. Additionally, a full physical examination, including abnormal signs since the last visit, and application site abnormality, was performed at each visit by the investigator. Global assessment of efficacy, tolerance, and ease of use (GAS, 1 = very poor to 5 = excellent) was assessed on day 28 by the owner, independently of the investigator (Appendix A). PS, IPS, and GAS have previously been used internally for other clinical studies, although not published in dogs.

The investigator applied the product on ten dogs on day 0 and day 14. One Effitix^®^ spot-on (Virbac, Carros, France) pipette was used per dog on each day of treatment [XS (26.8 mg fipronil, 240 mg permethrin) for dogs between 1.5–4 kg; S (67 mg fipronil, 600 mg permethrin) for dogs 4–10 kg; M (134 mg fipronil, 1200 mg permethrin) for dogs 10–20 kg; L (268 mg fipronil, 2400 mg permethrin) for dogs 20–40 kg; or XL (402 mg fipronil, 3600 mg permethrin) for dogs 40–60 kg]. One drop was applied to each affected site. The rest was applied to the skin between the scapulae. Five dogs were left untreated.

The variable was the resolution of the harvest mites infestation. PS reduction was calculated at each time point *t* using the arithmetic mean of PS according to the Abbott formula: PS reduction (%) = 100 × (mean day 0 − mean day *t*)/mean day 0. Skin lesion score reduction and IPS reduction were calculated using the same formula.

Scores obtained on days 1, 14, and 28 were compared with the baseline obtained on day 0 using Wilcoxon signed-rank tests. Continuity correction was applied. Ranked ex-aequos were detected, and appropriate corrections were applied. Significance was defined as *p* < 0.05. XLSTAT 2017-02 (Addinsoft SARL, Paris, France) was used to perform statistical analyses.

## 3. Results

### 3.1. Animal Population

All 15 dogs completed the study. There were 9 females and 6 males aged between 1.5 and 10 years (median = 6) and weighing between 5 and 23 kg (median = 15). There were various breeds (Brittany dogs (3), beagles (2), cocker spaniels (2), labrador crossed (2), Yorkshire terriers (2), whippets (2), fox terrier (1) and Staffordshire bull terrier (1)). Further to the application of the product, no adverse effect was noted.

### 3.2. Parasite Score

Parasitic scores (PS) of the control dogs were stable throughout the study (mean ± SD): 1.8 ± 0.8 on days 0 and 1, 2.2 ± 0.4 on day 14, and 1.8 ± 0.4 on day 28.

On day 0, treated dogs had a PS of 2.4 ± 0.7. On day 1, a 79% reduction in PS was observed (*p* < 0.002) and five dogs were free of mites; on days 14 and 28, the PS reduction was 92% and 96% (*p* < 0.002), respectively. On days 14 and 28, nine dogs out of 10 were parasite-free (Figure 1).

### 3.3. Other Parameters

The lesion score and investigator pruritus score (IPS) in control dogs remained steady. At inclusion, treated dogs had a lesion score of 2.9 ± 0.9. The score was unchanged on day 1. It was significantly reduced by 79% and 97% on days 14 and 28, respectively, (Table 1).

The inclusion investigator pruritus score (IPS) was 2.2 ± 0.8. It was slightly reduced on day 1, and subsequently significantly reduced on day 14 (−82%) and day 28 (−91%).

An average global assessment of efficacy (GAS) of 4.8 ± 0.4 (min 4, max 5) was given by the owners.

## 4. Discussion

Control and prevention of harvest mites are known to be difficult due to the parasite life cycle [3]. In addition to the miticidal effect, repellency can be considered as important both from an epidemiological standpoint, by preventing the transmission of pathogens [9,10,11], and also from a clinical perspective, by preventing potential irritation caused by chiggers [1]. The combination of fipronil and permethrin has repellent and insecticidal/acaricidal effects and represents an effective strategy for reducing the risk of transmission of zoonotic pathogens [14]. The repellent effect of the combination is due to permethrin [14]. The product is licensed in dogs to be used against infestations with fleas and/or ticks and when repellent activity is also necessary against sand-flies and/or mosquitoes. The product has a persistent insecticidal/acaricidal efficacy for up to 4 weeks against fleas [15], ticks [16,17], sandflies [18], and mosquitoes [19]. Very recently, the product was successfully evaluated on adult triatomines (*Rhodnius prolixus*) in dogs [20].

These results provide additional information to the previous studies that evaluated the efficacy of a spray of fipronil against harvest mites, [12] and the efficacy of permethrin in a line-on or a spray formulation [13]. In the first study, the authors used fipronil spray applied to the whole body to treat 18 dogs and three cats once monthly from diagnosis until the end of the season, without environmental restriction. Mites were eliminated without recurrence in 15 out of 18 dogs within one month of treatment. Two dogs had mites on their feet 14 days after the first treatment and received additional local treatment, whereas one dog did not benefit from treatment, possibly due to poor client compliance [12]. In the second study, spot-on and spray formulations of permethrin and pyriproxyfen were used in 15 naturally infested dogs in the North of France. Parasites were no longer detected three weeks after treatment in 14 out of 15 dogs, whereas low numbers of parasites persisted in the remaining dog. Four dogs required a second application of the treatment after 14 days [13].

The fipronil and permethrin ectoparasiticide combination which was evaluated in the current study offers very good protection (i.e., >90%) against harvest mites in dogs for two weeks following a topical application. As in a comparable study with a fipronil spot-on applied on cats [21], the spot-on solution was directly applied on the infested sites and a control group was included. The application was facilitated by the design of the pipette, allowing the precise control of the applied volume by a gentle pressure on the body of the pipette, resulting in a drop by drop application. The label of the permethrin-fipronil spot-on formulation recommends a topical application in two to four spots on the back from the shoulder to the base of the tail, depending on the size of the dog. This type of formulation allows the product to spread on the skin from the application spots and the concentration is expected to be minimal at the most distanced body regions including ears and interdigital spaces. We therefore purposely followed an off-label application of the product: direct application on infested body regions to reach a higher concentration of the product where the parasites attach. Twenty-four hours after the initial application, the efficacy was 79%, with five of the ten treated dogs who were free of mites. On day 14, only one dog displayed mites at a low infestation level. Few mites were also present on the same dog on day 28. This could be possibly explained either by a high parasitic pressure in the environment and/or an excessive licking of the interdigital area due to pre-existing pruritus, where the product had been applied.

In the few reports investigating activity against harvest mites, none had a control group [12,13,22]. We elected to include a control group to ensure that the reduction of the lesions in dogs receiving the treatment was not spontaneous and avoid misinterpretation of the results as trombiculiosis is a seasonal skin disorder. The study was conducted during September and October, and depending on the years, in the study area, the first-night frosts in the autumn can occur by the end of September. As a consequence, harvest mites population could decrease. Therefore, five dogs living in different households, but in the same geographical areas, were included in parallel and left untreated, to ensure that the tested population would be exposed to potential re-infestations all along the study.

In the present report, the product was applied twice at a two-week interval, based on published results [12,13,14]. All dogs tolerated the two applications very well. The antiparasitic efficacy was accompanied by a significant improvement of the lesional score at days 14 (−79%) and 28 (−97%). Understandably, the lesions were still present the day following the first application (day 1) despite a dramatic decrease of the PS (−79%). Similarly, the severity of pruritus was significantly decreased on day 14 (−82%) and day 28 (−91%), but it was almost unchanged on day 1 (−9%).

## 5. Conclusions

In these dogs with localized harvest mites infestation, the permethrin-fipronil combination was effective and well-tolerated. The product should be recommended every two weeks in the fall in dogs who suffer from trombiculiosis. It should be emphasized that this is the first clinical study in dogs with harvest mites that has been done with a control group. Furthermore, the topical application on lesion sites is likely enhancing the efficacy.

## Figures and Tables

**Figure 1 vetsci-06-00100-f001:**
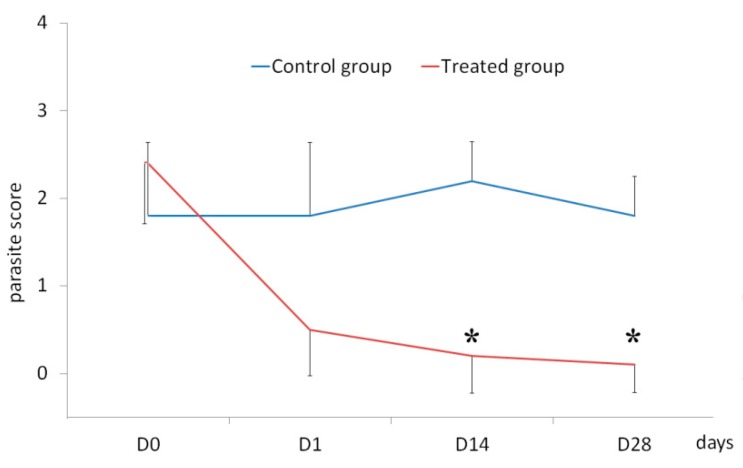
Line graph of the progression of the arithmetic mean ± SD of parasite scores in control dogs (blue) and treated dogs (red) infested by *Neotrombicula autumnalis* at days 0, 1, 14, and 28. Treated dogs received a permethrin 54.5% and fipronil 6.1% based spot-on solution (Effitix^®^; Virbac, Carros, France) on day 0 and day 14. * Significant difference from baseline within a group of dogs (*p* < 0.05).

**Table 1 vetsci-06-00100-t001:** Lesion score, investigator pruritus score and global assessment of efficacy at days 0, 1, 14, and 28 of treated dogs and control dogs infested by *Neotrombicula autumnalis*.

	Lesion Score				IPS				GAS
	D0	D1	D14	D28	D0	D1	D14	D28	D28
Treated dogs ^1^	2.9 ± 0.9	2.9 ± 0.9	0.6 ± 0.5	0.1 ± 0.3	2.2 ± 0.8	2 ± 0.7	0.4 ± 0.5	0.2 ± 0.4	4.8 ± 0.4
Control dogs	0.4 ± 0.5	0.4 ± 0.5	0 ± 0	0.2 ± 0.4	0.6 ± 0.9	0.6 ± 0.9	0.4 ± 0.9	0.4 ± 0.5	/
Percentage of reduction/D0 ^2^	/	0%	79%	97%	/	9%	82%	91%	/
*p*-value	/	/	0.002	0.002	/	0.5	0.004	0.002	/

^1^ Treated dogs received a permethrin 54.5% and fipronil 6.1% based spot-on solution (Effitix^®^; Virbac, Carros, France) on day 0 and day 14. IPS: investigator pruritus score; GAS: global assessment of efficacy. ^2^ Percentage of reduction compared to D0.

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
