# Peer review of "Treatment of Harvest Mite Infestation in Dogs Using a Permethrin 54.5% and Fipronil 6.1% (Effitix®) Topical Spot-On Formulation"

_vetsci, 2019, doi:10.3390/vetsci6040100_

Round 1

Reviewer 1 Report

The paper might be published, since there the efficacy of the tested products is shown viceversa a control group when undergone exposition to harvest mites, while several similar tests often had no control group.

Author Response

The paper might be published, since there the efficacy of the tested products is shown viceversa a control group when undergone exposition to harvest mites, while several similar tests often had no control group.

Thank you very much

Reviewer 2 Report

Lecru et al. Treatment of harvest mite infestation in dogs using permethrin 54.5% and fipronil 6.1% topical application on lesion sites.  

Abstract: if word limit allows, I would suggest avoiding combining the data from each day and variable into one long sentence, as it is hard to read as it is.  

Introduction: 

Line 31: Any reason why this sentence is its own paragraph?  

Line 33: Attacking seems an extreme word. Would consider finding another one.  

Line 35: 'highest numbers' in nature? on the host? Do you mean 'burden'?  This sentence is confusing.  

Line 36: I would move this part to be right after the life cycle later.

Line 37: If it's one species, consider writing the sentence as singular noun.  

Line 39: consider deleting section on the other two mite species; they seem out of place here.  

41: 0.7mm diameter? long? what shape are the mites?

43: they molt to nymphs but mature to adults? do they molt into adults?

line 45: southern hemisphere? southern part of the northern hemisphere? please be more specific.

line 54: probably a language difference, but use '2,000 larvae'

line 54: I'm not sure what a 'nervous sign' is?

line 56: haven't read the reference, but ensure that these were simply detected (probably by PCR) and not truly 'isolated' 

Methods:

line 70: which university?

line 73: are multiple species of mites potentially involved? was mite identification done? any other specific diagnostics to ensure they were harvest mites and of which species is important.  

lines 86-90: although seemingly obvious, specify that these lesions were associated with mites rather than could be from some other skin pathogen.  

line 93-94: IPS 1 and 2 sound like the same to me.

lines 91-95: I think the extent of grooming and agitated is fair; determining 'uncomfortable' is very subjective and could just be because they were in the clinic making them nervous.  

were controlled dogs also given an IPS?

Lines 99-100: There needs to be more detail on the GAS; what it means, how it is determined, etc.  If it is a questionairre, maybe include as a supplemental file.  

Line 112: Due to the number of tests, I would've considered some sort of correction, such as a Bonferroni.  

Results:

Figure: These boxplots are unlike any i've seen with days 0 and 1 having a solid-filled area and 14 and 28 do not.  The description (lines 133-136 mentions a solid line within the box, but the last four lines do not have a box.  I'm not sure I see a huge value in including the mean and the median; it makes the figure a little busy.  Also, a line graph in some ways would show this data better than boxplots.    

Lines 133-136 should be in the caption, not in the text.  Also, the figure caption claims this is parasite score, but lines 133-136 say that it is the lesion score.  

No figures on the other parameters? 

A single table for each variable at each time frame would be worthwhile.  As it stands, it is difficult to read in the text, and this approach could replace adding figures for the other variables, which I think would improve the manuscript.  

Discussion:

Line 150: month "of" treatment

Line 187: all dogs tolerated the two applications very well.

Line 192-194: this sentence seems out of place.  

Overall: 

The discussion mentions studies that have looked at similar compounds and the efficacy against harvest mites.  I would emphasize in the introduction or discussion what specifically makes this study unique (use of controls, better efficacy if used in targeted areas vs on the back, etc.).  The data here are likely useful, but the results were difficult to interpret, which made comparing the data to other studies/compounds challenging.  If any photos were taken to show the range of severity of each variable, that would also be interesting to see and provide visually that there is real improvement within each patient post treatment.  

Reviewer 3 Report

Dear authors, Thanks for your efforts. I have some few comments. 1- In title: "topical application on lesion sites" should be omitted. As you sue on site and on shoulder which is not on sites. 2- In Abstract: P1, L 17: Ten naturally infected dogs 3- In Materials and methods: P2, L 73-79: How did you detect and confirm the mite species infect dogs used in this study? 4- In Materials and methods: P2, L 83: "Parasite Score" what parasite score exactly you mean? 5- In Materials and methods: P2, L 83: How did you detect and count mites by using magnifying lens?!! if there are a reference for this point, cite it please. 6- In Results section: lesion sites were not clear?! are all lesions in all dogs were in the same site or different sites of their body? 7- In discussion section: you discuses your results in a good manner, but you did not show the drawbacks or contraindications of using such combination of active principles?, you should discus more about the active principles (mode of actions, synergistic effect, contraindications...etc) Thanks. Regards

Round 2

Reviewer 2 Report

Lecru et al. treatment of harvest mites in dogs.  

This version of the manuscript contains major improvements, particularly in the presentation of results.  I have three very minor comments.

1:  The paragraph formatting is a little distracting.  The introduction contains one very long paragraph, the discussion contains a paragraph with only one sentence, and the discussion has spaces between paragraphs, which is not consistent with the other sections.

2:  Figure 1 at D0, it is difficult to tell which which group the SD bar is related to.  Could both bars be added but slightly offset so that they can be seen?

3:  Line 213: do you mean effective rather than efficient?  

Reviewer 3 Report

Dear authors, 

Thanks for your efforts .

Regards. 

Author Response

We understand that reviewer 3 is not completely satisfied with the revised manuscript which was submitted. However, no comment, suggestion or query is indicated by the reviewer.

Hence we did not modified the manuscript.